# Conditional Knockout of Neurexins Alters the Contribution of Calcium Channel Subtypes to Presynaptic Ca^2+^ Influx

**DOI:** 10.3390/cells13110981

**Published:** 2024-06-05

**Authors:** Johannes Brockhaus, Iris Kahl, Mohiuddin Ahmad, Daniele Repetto, Carsten Reissner, Markus Missler

**Affiliations:** 1Institute of Anatomy and Molecular Neurobiology, University of Münster, 48149 Münster, Germany; 2Department of Cell Biology, College of Medicine, University of Oklahoma Health Sciences Center, Oklahoma City, OK 73104, USA

**Keywords:** neurexin, calcium channel subtypes, presynapse, endocannabinoid system, genetically encoded calcium indicator

## Abstract

Presynaptic Ca^2+^ influx through voltage-gated Ca^2+^ channels (VGCCs) is a key signal for synaptic vesicle release. Synaptic neurexins can partially determine the strength of transmission by regulating VGCCs. However, it is unknown whether neurexins modulate Ca^2+^ influx via all VGCC subtypes similarly. Here, we performed live cell imaging of synaptic boutons from primary hippocampal neurons with a Ca^2+^ indicator. We used the expression of inactive and active Cre recombinase to compare control to conditional knockout neurons lacking either all or selected neurexin variants. We found that reduced total presynaptic Ca^2+^ transients caused by the deletion of all neurexins were primarily due to the reduced contribution of P/Q-type VGCCs. The deletion of neurexin1α alone also reduced the total presynaptic Ca^2+^ influx but increased Ca^2+^ influx via N-type VGCCs. Moreover, we tested whether the decrease in Ca^2+^ influx induced by activation of cannabinoid receptor 1 (CB1-receptor) is modulated by neurexins. Unlike earlier observations emphasizing a role for β-neurexins, we found that the decrease in presynaptic Ca^2+^ transients induced by CB1-receptor activation depended more strongly on the presence of α-neurexins in hippocampal neurons. Together, our results suggest that neurexins have unique roles in the modulation of presynaptic Ca^2+^ influx through VGCC subtypes and that different neurexin variants may affect specific VGCCs.

## 1. Introduction

Synaptic transmission is a fundamental step in neuronal communication and the main place for neuromodulation. In presynaptic boutons, the opening of high-threshold voltage-gated calcium channels (VGCCs) is a central step in the action-potential-driven transmitter release [1,2]. Synaptic strength and synchronous release depend on the subtype, number, activity, and topography of VGCCs [3,4]. Action-potential-triggered vesicle release mainly depends on Cav2.1 (P/Q-type) and Cav2.2 (N-type) VGCCs as determined by postsynaptic excitatory postsynaptic currents (EPSCs) [5,6], but in some synapses, only P/Q-type VGCCs seem to be relevant for fast synaptic vesicle release [7]. Additional Ca^2+^ influx at the presynapse employs Cav2.3 (R-type) and Cav1.2/3 (L-type) channels [8,9], although the latter are believed to have a limited impact on vesicle release and, thus, eIPSC amplitude [1,7,10]. The high-voltage-activated Cav2 channels show faster activation and inactivation making them suitable for fast transmission of neuronal action potential activity, whereas Cav1 channels are primarily involved in slower processes like hormone secretion and Ca^2+^ signaling to gene transcription [11]. Accordingly, biochemical and functional studies have identified numerous molecular interactions between VGCC subunits and various partners that serve, for example, to couple Ca^2+^ channels to the release machinery (reviewed in [11,12]). Interestingly, these interactions of VGCC subunits include not only intracellular pathways but also crosstalk to extracellular or cell surface molecules [13,14,15,16,17,18,19,20].

We discovered many years ago that neurexins (Nxs), a polymorphic family of synaptic cell surface molecules [21,22], are involved in the regulation of VGCC-dependent neurotransmitter release from excitatory and inhibitory synapses [23]. Neurexins are encoded by three genes in vertebrates, each of which contains independent promoters that drive transcription of longer α-Nxs and shorter β-Nxs. A truncated γ-isoform is transcribed in neurexin-1 [24], and more variants arise from up to six conserved splice sites [25,26]. Extracellularly, α-Nx proteins mostly comprise six laminin-Nx-sex-hormone-binding (LNS) domains with interspersed epidermal growth factor (EGF)-like repeats. Shorter β-Nxs differ by expressing a β-specific, 37-residue-long N-terminal domain before splicing into the last (sixth) LNS domain of the respective gene [21,22]. Since LNS6 and subsequent sequences are identical in α- and β-Nxs, they share properties such as a C-terminal PDZ recognition motif required for intracellular trafficking [27,28], a heparan sulfate glycan moiety [29], and physiological ectodomain cleavage [30]. α- and β-Nxs also share binding partners such as neuroligins [31,32,33], leucine-rich repeat transmembrane neuronal proteins (LRRTMs) [34,35,36], α-dystroglycan [37,38], latrophilins [39], and cerebellins [40,41].

The functional link between Nxs and VGCCs was initially observed in a constitutive deletion mouse model (knockout) of all α-Nxs [23,42,43] and later confirmed in conditional knockout neurons lacking all β-Nxs [44,45]. Surprisingly, investigations of conditional knockout neurons lacking all Nx variants detected reduced total Ca^2+^ transients only in somatostatin- but not parvalbumin-positive interneurons of the medial prefrontal cortex [46] and failed to see reduced Ca^2+^ influx into the parvalbumin-positive excitatory calyx of Held synapses in the brainstem [47]. A possible explanation for this discrepancy might be that the functional link between Nxs and VGCCs involves specific combinations of Nx variants and VGCC subtypes which may differ between brain regions and subpopulations of synapses. In support, we found recently that the reduced Ca^2+^ influx into boutons of excitatory hippocampal neurons in α-Nx triple-knockout mice predominantly involved Cav2.1 (P/Q-type) VGCCs [17] and could be rescued by overexpression of the Nx1α variant which is abundant in hippocampal neurons [48].

To further explore this important aspect in our current study, we directly compared whether and how deletions of one or all Nx isoforms can affect different synaptic VGCC subtypes in the same model system. Therefore, we generated a conditional Nx1α knockout mouse model and compared presynaptic Ca^2+^ influx in primary hippocampal cultures of control to conditional knockout neurons lacking either the single Nx1α variant (Nx1α cKO, created for this study) or all Nx isoforms (Nx123 cKO [46,47]). We particularly focused on how the deletions affect single-action-potential-evoked Ca^2+^ influx through different VGCC subtypes, using transfected synGCaMP7b [49] as a Ca^2+^ indicator and pharmacological isolation by sequential addition of subtype-specific blockers [9] which together allowed quantification even at the level of individual presynaptic boutons. We report here that Nx variants likely alter the contribution of most VGCC subtypes to presynaptic Ca^2+^ transients, including P/Q-type (CaV2.1), N-type (CaV2.1), L-type (CaV1.2/3), and R-type (CaV2.3) channels. Strikingly, the deletions of a single Nx1α or all Nx variants resulted in a different pattern of VGCC subtypes affected. These findings may indicate that Nx variants modulate Ca^2+^ influx in a partially overlapping, partially unique way, depending on the actual presence and/or relative amount of Nx variants and VGCC subtypes in a particular synapse population or even in individual terminals.

## 2. Materials and Methods

### 2.1. Animals

Mice of either sex were used for neuronal cultures derived from timed-pregnant females at E17. Animal experiments were performed at the University of Münster following government regulations for animal welfare and approved by the Landesamt für Natur, Umwelt und Verbraucherschutz (LANUV, Düsseldorf, NRW, Germany), license numbers 84-02.05.20.11.209 and 84-02.04.2015.A423.

Three mouse models were used in this study, Nx1α cKO, Nx123 cKO, and β-Nx cKO. Nx123 cKO and β-Nx cKO were characterized and reported earlier [44,45,46]. The conditional knockout model for Nx1α (Nx1α cKO) is reported here for the first time. Briefly, a targeting vector was cloned to introduce loxP sites on either side of the first coding exon of the Nx1 gene based on mouse genomic clones. This exon is the largest of all the exons of the gene and codes for the signal peptide as well as the first LNS domain and EGF domain. It was expected that the deletion of this exon would lead to a complete loss of functional protein from the synapses because the same exon was deleted to generate a conventional knockout of Nx1α [50] and led to the loss of the complete protein [23,50]. 5′ and 3′ loxP sites were inserted along with selection markers (Figure 1B), and the targeting vector was electroporated into ES cells. Homologous recombination was identified by Southern blotting and PCRs; positive ES cell clones were microinjected into blastocysts. Chimeric mice were generated, and germline transmission was monitored again by Southern blotting and PCRs, as described before [23,50]. Homozygous KI mice are viable and can be kept on a homozygous background.

### 2.2. Neuronal Cell Culture

Dissociated primary neurons were prepared in Hank’s Balanced Salt Solution (HBSS) from hippocampi as described [28,45]. Briefly, cell suspensions obtained after 0.25% trypsin and trituration were plated onto 18 mm glass coverslips coated with poly-L-lysine (Sigma) at a density of 40,000 cells/coverslip. After 4 h at 37 °C in plating medium (MEM, 10% horse serum, 0.6% glucose, 1 mM sodium pyruvate), coverslips were inverted onto a 70–80% confluent monolayer of astrocytes grown in 12-well plates (Falcon) and incubated in Neurobasal medium supplemented with B27, 0.5 mM glutamine, and 12.5 mM glutamate. After 3 days, media were refreshed with Neurobasal medium supplemented with B27, 0.5 mM glutamine, and 5 mM AraC. Cultures were maintained at 37 °C in a humidified incubator with an atmosphere of 95% air and 5% CO_2_. Neurons were transfected at day in vitro (DIV) 11 using lipofectamine (Thermo Fisher Scientific, Waltham, MA, USA), and experiments were performed between DIV 17 and DIV 21.

For induction of the conditional knockout of Nx genes marked with loxP sites, neuronal cultures were infected at DIV 4 with lentivirus by adding 100 µL per well of viral supernatant that was made as described earlier [45]. In short, recombinant lentiviral particles were produced in HEK293 cells, and the supernatant was collected and snap-frozen (−80 °C). The lentivirus contained EGFP fused to active Cre recombinase (Cre), or to an inactive mutated Cre recombinase (Cre^mut^) [45,51], or the same vector with Cre recombinase deleted (ΔCre) [52].

### 2.3. Ca^2+^ Imaging

For Ca^2+^ imaging of synaptic boutons using a genetically encoded indicator, we generated the expression plasmid synGCaMP7b by fusing GCaMP7b [49] to synaptophysin, driven by a human synapsin promotor as described and characterized earlier for synGCaMP6f [9,17,45].

To determine presynaptic Ca^2+^ influx, primary neurons were transfected at DIV 11 with synGCaMP7b (see above) and co-transfected with pMH4-SYN-tdimer2-RFP (RFP, T. Oertner, Hamburg, Germany) for better identification of neuronal morphology. As only a few neurons were transfected by lipofectamine (about 30–50 per coverslip), it was possible to observe areas where only presynaptic boutons were loaded with synGCaMP7b (Figure 2A,B). Six to eight days post-transfection, neurons growing on glass coverslips were placed in a recording chamber mounted to an inverted microscope (Observer.A1, Zeiss, Oberkochen, Germany) in 2 mL bath solution (temperature 32 °C), containing (in mM) NaCl 145, KCl 3, MgCl_2_ 1, CaCl_2_ 2, glucose 11, and HEPES 10, with pH 7.3 adjusted with NaOH; to suppress postsynaptic signaling, 10 μM 6-cyano-7-nitroquinoxaline-2,3-dione (CNQX), 25 μM D, L-2-amino-5-phosphonovaleric acid (AP5), and 10 μM bicuculline were added. All chemicals were obtained from Sigma (St. Louis, MO, USA), except Ca^2+^ channel blockers (Alomone Labs, Jerusalem, Israel). A stimulation electrode, built by two platinum wires of 10 mm length in 10 mm distance was positioned with a micromanipulator (MPC-200, Sutter Instrument, Novato, CA, USA), and neurons were stimulated with 50 Hz trains of 1, 3, or 10 current pulses (1 ms, 55 mA). Ca^2+^ transients were visualized and recorded (20 ms exposure time, frame rate 50 Hz, binning 2: 0.46 μm per pixel) with a CMOS camera (Orca Flash4.0, Hamamatsu, Japan) and an LED light source (SpectraX, Lumencor, Beaverton, OR, USA) using the green channel (excitation at 470 ± 20 nm) or red channel (640 ± 20 nm) and controlled by VisiView 4.0 software (Visitron Systems, Puchheim, Germany). As a standard, 20 frames were recorded before the stimulus train was triggered. For stimulation with one AP, four individual recordings with 10 s time intervals were averaged frame by frame to improve the signal-to-noise ratio. Ca^2+^ channel antagonists were added by direct application into the recording chamber 3 min before the next stimulation: ω-agatoxin IVA (0.1 μM, P/Q-Type; Alomone Labs), ω-conotoxin GVIA (2 μM, N-Type; Alomone Labs), nifedipine (20 μM, L-Type; Sigma-Aldrich, St. Louis, MO, USA), and SNX-482 (0.5 μM, R-Type; Alomone Labs). Due to the chemical instability of 2-AG (2 μM, CB1-receptor agonist; Avantilipids, Alabaster, AL, USA) in aqueous solutions [53], 1 μL aliquots of a 4 mM 2-AG in DMSO were prepared and frozen at −20 °C. Immediately before use, 200 μL of bath solution was added and inserted into the recording chamber.

### 2.4. Data Analysis

Data analysis of imaging recordings of Ca^2+^ transients was performed with Fiji/ImageJ 1.520 (National Institute of Health, MA, USA) and IgorPro 6.3 (Wavemetrics, Lake Oswego, Oregon) or MATLAB R2020b (The MathWorks Inc., Natick, MA, USA). Active boutons were identified by the increase in fluorescence (ΔF) after stimulation with a train of 3 APs. The plugin “Time Series Analyzer V3” was used to draw 60–100 regions of interest (ROIs) per measurement around active boutons with an AutoROI diameter of 8 pixels (Figure 2C). To quantify fluorescence changes in individual boutons, we first applied the commonly used [54] “Subtract Background...” tool of Fiji/ImageJ (employing a “rolling ball” algorithm with a radius of 20 pixels ~ 10 μm), to remove the background signal deriving from faint autofluorescence and the dark current of the camera. For each ROI and each frame, the mean of the four pixels with the strongest fluorescence was calculated using a self-made macro. The area of four pixels (0.85 mm^2^) corresponds to the size of a normal bouton, and the restriction to the four brightest pixels avoids the problem of the relevance of the ROI size to the area of increased fluorescence. Further calculations used IgorPro or MATLAB to average for each ROI the value of frames 10–20 as a control value (F_0_); changes were calculated as the change in fluorescence intensity (F_stim_ − F_0_ = ΔF) divided by the control (ΔF/F_0_) for each ROI. Single-AP responses were analyzed after averaging four consecutive recordings already within ImageJ, and for the analysis of individual amplitudes, the traces were binomial Gaussian smoothed (coefficient 3) to improve the signal-to-noise ratio.

To quantify the relative change caused by the application of the CB1-receptor agonist 2-AG, the relative change in synGCaMP7b ΔF/F_0_ was calculated for each bouton as described in the following formula:$$rel.change\left(\%\right)=-\left(1-\frac{{\left(\Delta F/F_{0}\right)}_{post}}{{\left(\Delta F/F_{0}\right)}_{pre}}\right)\times 100$$

For the analysis of individual presynaptic boutons, only boutons with an amplitude larger than 0.12 ΔF/F_0_ (three times the noise level) before treatment were included to allow reliable quantification of minor reduction.

### 2.5. Quantification and Statistical Analysis

Statistical tests were performed in Prism (GraphPad Prism 6.0d, GraphPad Software Inc., Boston, MA, USA). If data showed a normal distribution, Student’s *t*-test was used to compare two groups, and ANOVA was used for multiple comparisons with post hoc Turkey’s multiple-comparison test. In case the criteria for normal distribution were not fulfilled, the corresponding non-parametric tests were used, e.g., Kruskal-Wallis test followed by Dunn’s test. The data are represented as mean ± SEM or boxplot with 25–75 percentile and significance level indicated by asterisks (* *p* < 0.05, ** *p* < 0.01, *** *p* < 0.001, and **** *p* < 0.0001). Further information on statistical details can be found in the figure legends. The experiments were not randomized, and investigators were only partially blinded during experiments and analyses. For outlier identification (see figure legend), the ROUT method [55] was performed in GraphPad Prism with Q = 1. In addition, the numbers of examined neurons and boutons are shown in the form of boutons/neurons or as a single number indicating the number of neurons in all figures.

## 3. Results

### 3.1. Nx1α Is the Prominent Nx Variant in Cultured Primary Hippocampal Neurons

Cultured primary hippocampal neurons from Nx123 conditional knockout (cKO) mice [46] were transduced by lentivirus particles expressing active Cre- or inactive ΔCre-recombinase fused to EGFP at DIV4. Neurons were tested for α-Nx expression at DIV18 by immunoblotting and compared to neurons that contained no floxed α-Nx variants (Figure 1A). All three antibodies tested (Figure 1A_1_–A_3_) revealed the presence of α-Nx in wild-type and ΔCre-transduced neurons, and strongly reduced protein levels in Cre-transduced Nx123 cKO neurons, indicating an effective deletion of α-Nxs. This finding is consistent with the reduced mRNA levels in the same mouse model [46] and the efficient removal of β-Nx protein upon Cre recombination in a related line [45]. To compare these Nx123 cKO neurons to neurons lacking only the single Nx1α variant, which is prominently expressed in the hippocampus [48], a knock-in/conditional knockout mouse line of Nx1α was generated, in which the first coding exon of the neurexin-1 gene is flanked by loxP sites (Figure 1B). In hippocampal neurons cultured from these Nx1α cKO mice, Cre recombinase expression also efficiently depleted the α-Nx signal on immunoblots in comparison to cultures infected by ΔCre-expressing lentivirus (Figure 1C_2_) in contrast to neurons that contained no floxed α-Nxs (Figure 1C_1_). The strong reduction in the signal indicated an effective deletion of Nx1α. Moreover, since we used an antibody that recognizes all α-Nx variants, this finding confirms the substantial contribution of the Nx1α isoform to the overall α-Nx protein pool, consistent with earlier mRNA data [48]. The strong reduction in the signal also indicates that the deletion of Nx1α is likely not compensated by other α-Nx isoforms. Quantification of the Nx signal intensities on repeated immunoblots, normalized to the respective control (ΔCre) value (Figure 1D), confirmed the predominance of Nx1α in the cultured hippocampal neurons because the deletion of the single variant reduced the α-Nx signal already by 75% to 25 ± 2% of control (Nx1α cKO Cre), while the deletion of all α-Nxs reduced the signal to 8 ± 3% (Nx123 cKO Cre).

### 3.2. Deleting the Single Nx1α Variant Is Sufficient to Reduce the Total Presynaptic Ca^2+^ Influx

To analyze single-action-potential-driven presynaptic Ca^2+^ influx, we measured Ca^2+^ transients using the genetically encoded Ca^2+^ indicator synGCaMP7b (Figure 2A–C). We then compared control neurons (Cre^mut^ in Figure 2D,E) to Nx-deficient neurons (Cre in Figure 2D,E). In hippocampal neurons lacking all Nx variants, the maximum amplitude of Ca^2+^ transients was 0.33 ± 0.02 ΔF/F_0_ (Figure 2F, Nx123 cKO Cre) compared to 0.43 ± 0.02 ΔF/F_0_ in neurons with normal Nx expression (Figure 2F, Nx123 cKO Cre^mut^; *p* < 0.0001, unpaired *t*-test). This corresponds to a reduction by 23.3% in Nx123 cKO Cre compared to Nx123 cKO Cre^mut^ neurons, which is compatible with the slightly smaller reduction in total Ca^2+^ transients (18.5%) we found earlier in constitutive KO neurons that lack all α-Nxs [17].

**Figure 2 cells-13-00981-f002:**
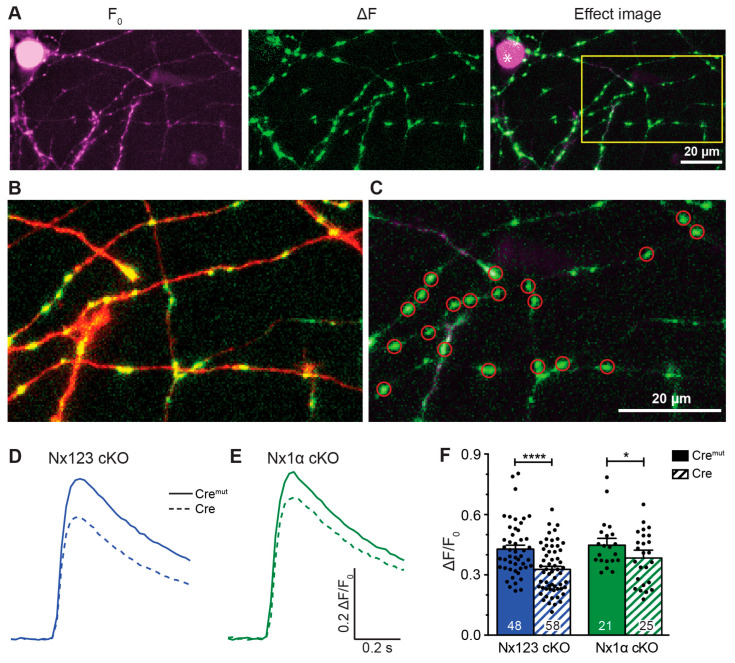
Presynaptic Ca^2+^ transients recorded from individual active boutons with synGCaMP7b. (**A**) Example picture of fluorescence intensity of synGCaMP7b before stimulation (left, F_0_, shown in magenta), representing the baseline fluorescence; fluorescence intensity changes after stimulation with 3 APs, isolated by subtraction (middle, ΔF, shown in green). The green fluorescence dots lighting up indicate active boutons. Both images merged represent the effect image (right) that allows the identification of active boutons that are not disturbed by high baseline fluorescence of other sources like Cre-EGFP-fluorescent cell nuclei (asterisk). (**B**) Enlarged perspective (yellow box in **A**), showing the change in fluorescence (ΔF, green) as well as the cell process morphology indicated by co-transfected RFP (red). (**C**) ROIs (red circles) were placed on active boutons for the quantification of presynaptic Ca^2+^ transients. (**D**) Averaged synGCaMP7b fluorescence changes from Nx123 cKO neurons with Nxs (Cre^mut^, n = 14 cells/1045 boutons) or without all Nx variants (dashed line, Cre, 13/916) show Ca^2+^ transients following a single-AP stimulation. (**E**) Neurons lacking only Nx1α (Cre, 8/681) and equivalent controls (Cre^mut^, 12/1074) showed comparable fluorescence alterations as those seen in N123 cKO. (**F**) Comparing peak values of Ca^2+^ transients (mean ± SEM) in Cre^mut^ and Cre cells from both mouse lines in response to a single-AP stimulation. Nx123 cKO: Cre^mut^ 48 cells/3964 boutons, Cre 58/4582; Nx1α cKO: Cre^mut^ 21/1672, Cre 25/2051. The mean values of all boutons of a single cell are shown as dots and used for statistics. Columns were compared with an unpaired *t*-test. * *p* < 0.05, **** *p* < 0.0001.

These results suggest that the α-Nx variants are predominantly responsible for regulating the total presynaptic Ca^2+^ influx. To test if deletion of the single Nx1α variant was already sufficient to reduce the total presynaptic Ca^2+^ influx, we performed the same experiment using neurons from the newly generated Nx1α cKO mouse line (Figure 2E,F). We found that again the maximum amplitude of Ca^2+^ transients was smaller in Cre-transduced neurons compared to the control (Nx1α cKO, Cre^mut^: 0.458 ± 0.026 vs. Cre: 0.380 ± 0.025; *p* < 0.05; Figure 2F). This reduction in Ca^2+^ transients by 17.0% in neurons lacking only α-Nx is in line with the strong expression of the Nx1α isoform in hippocampal neurons [48], and constitutive deletion of Nx1α has been shown before to cause functional and behavioral deficits [56,57,58]. These data may indicate that the lack of Nx1α is so fundamental that other α-Nx variants, for example, Nx3α [48], cannot fully compensate for the loss at the level of overall Ca^2+^ influx, emphasizing the role of Nx1α for the presence and function of presynaptic VGCCs. The question arises, however, of whether all VGCC subtypes contribute equally to the reduction in overall Ca^2+^ influx, or if the individual VGCC subtypes (P/Q, N, L, or R) contribute disproportionally to the effect.

### 3.3. Deletion of All Nxs Predominantly Reduced Ca^2+^ Influx through P/Q-Type VGCCs

The total Ca^2+^ influx into presynaptic terminals in response to single action potentials is composed of contributions from different VGCC subtypes, which can be inhibited by specific blockers. In our experiments, we blocked P/Q-type channels by 0.1 μM ω-agatoxin IVA, N-type channels by 2 μM ω-conotoxin GVIA, L-type channels by 20 μM nifedipine, and R-type channels by 0.5 μM SNX-482. Sequential administration of these blockers was used to isolate pharmacologically Ca^2+^ influx through individual subtypes, which we characterized before in our cell culture model [9]. In that previous study, we observed that sequential addition of the different VGCC blockers caused a reduction in Ca^2+^ influx after almost every addition, indicating a broad mixture of P/Q-type, N-type, L-type, and R-type VGCCs in presynaptic boutons of primary hippocampal neurons. We, therefore, applied the protocol of sequential blocker administration on Nx123 cKO neurons transduced by active Cre and inactive Cre^mut^ recombinase to dissect if deletion of all Nxs affected the presynaptic VGCC subtype composition or induced a proportional reduction in all subtypes.

We found in control neurons that P/Q-type VGCCs contributed most to Ca^2+^ transients, followed by L-type and N-type channels (Figure 3A, Cre^mut^). The contribution of R-type channels isolated by SNX-482 in normal boutons was so small that a reliable quantification in comparison to noise was hardly possible. The small Ca^2+^ transient that is still visible in the presence of all blockers is likely explained by some SNX-482-insensitive R-type channels [5,59,60]. More importantly, the reduced total presynaptic Ca^2+^ transients in neurons lacking all Nxs reported above (Figure 2) were mainly due to a substantial reduction in P/Q-type channel activity (Figure 3A, Cre), and additionally, the portion of N-type channels is moderately smaller, whereas some SNX-482-sensitive R-type channels could be identified here. The L-type channels seemed not affected by the loss of Nxs. The strong impact of Nxs on P/Q-type channels and the Nx indifference of L-type channels are visible in a direct comparison of the digitally isolated transients as shown in Figure 3B. The absolute Ca^2+^ influx through the different VGCC subtypes was quantified in more detail and compared between Nx-expressing (Cre^mut^) and Nx-deficient (Cre) neurons. The isolated Ca^2+^ transients passing through P/Q-type channels (Nx123 Cre^mut^: 0.17 ± 0.04 ΔF/F_0_) were reduced almost by half when all Nxs were missing (Nx123 Cre: 0.09 ± 0.03 ΔF/F_0_). Also, the N-type Ca^2+^ channel transients were reduced in Nx-deficient neurons, albeit at a lower level (Nx123 Cre^mut^: 0.080 ± 0.014 ΔF/F_0_ vs. Nx123 Cre: 0.057 ± 0.010 ΔF/F_0_), whereas the absolute contribution of L-type channels seems not affected by a lack of Nxs (Nx123 Cre^mut^: 0.148 ± 0.034 ΔF/F_0_ vs. Nx123 Cre: 0.143 ± 0.028 ΔF/F_0_). For the R-type Ca^2+^ channel, we observed no R-type transient in the presence of Nxs, whereas in the absence of Nxs, a small SNX-482-sensitive R-type transient was present (Nx123 Cre: 0.032 ± 0.009 ΔF/F_0_).

To compare the relative proportion of Ca^2+^ influx through the different VGCC subtypes, we calculated the share of VGCC subtypes to the total Ca^2+^ influx for individual boutons of Nx-expressing (Cre^mut^) and Nx-deficient (Cre) neurons (Figure 3C). As a consequence of the different total presynaptic Ca^2+^ influx of control and cKO boutons, the absolute Ca^2+^ influx through a particular VGCC subtype differs from their relative contribution. For example, the almost equal *absolute* Ca^2+^ influx via L-type channels (Figure 3B, green traces) corresponds to a larger *relative* contribution of L-type channels in N123 cKO (Figure 3C, green bars) as the *total* Ca^2+^ transient is smaller in N123 cKO. In control neurons, 41.2 ± 1.6% of the total Ca^2+^ influx passed through the P/Q-type Ca^2+^ channels, compared to 26.6 ± 1.5% in Nx-deficient cells (*p* < 0.0001; Kruskal-Wallis test; Figure 3C). In contrast to P/Q-type, the part relative to the total Ca^2+^ transient of the L-type Ca^2+^ channel influx was larger in Nx-deficient cells (Nx123 Cre^mut^: 29.9 ± 1.3% vs. Nx123 Cre: 38.6 ± 1.5%, *p* < 0.0001) without an increase in absolute amount (see Figure 3B) as the total Ca^2+^ transient was smaller in the Nx-deficient neurons. A lack of Nxs had only a minor impact on the relative portion of N-type channels (Nx123 Cre^mut^: 17.8 ± 0.9% vs. Nx123 Cre: 14.6 ± 0.9%, *p* = 0.1337). A small increase in Nx-deficient presynapses could be observed for R-type (Nx123 Cre^mut^: 1.6 ± 0.5% vs. Nx123 Cre: 7.2 ± 0.7%, *p* = 0.0028). Taken together, these results show that the VGCC subtype with the largest relative contribution shifted from P/Q-type channels in control conditions to the L-type channels in neurons lacking all Nxs.

The improved signal-to-noise ratio of the recordings with GCaMP7b allowed even an evaluation of VGCC subtype contribution not only on the cellular level but also in individual synaptic boutons. In these recordings, we observed a broad heterogeneity in the VGCC subtype contribution of individual synaptic boutons within the same neuron. For each bouton, the relative contributions of P/Q-, N-, and L-type channels to Ca^2+^ transients were calculated, and the frequency distribution was plotted as a histogram (Figure 3D–F). The analysis for the P/Q-type part in individual presynaptic boutons showed many boutons with a P/Q-type contribution of about 60–80% in control conditions, but only a few boutons with this amount of P/Q-type Ca^2+^ influx in neurons lacking Nxs indicated by a clear left-shift in the histogram with the maximum contribution of the P/Q-type in synaptic boutons lacking Nxs being around 20% (Figure 3D). N- or L-type channels had a maximum at 10–30% of Ca^2+^ influx in boutons with normal Nx levels, but the contribution reached above 90% within some boutons, indicating that in some individual boutons, the Ca^2+^ transients were driven almost completely by only one of these types of VGCCs. Regarding the deletion of all Nx variants, an altered distribution was observed. In Nx-deficient boutons, N- and P/Q-types revealed a similar contribution with a peak at 10–30%. In terms of the L-type, the contribution was more scattered, with most boutons having a contribution of about 50–60%. Consequently, it appeared that in Nx123 KO neurons, a shift took place in the opposite direction for P/Q-type (Figure 3D) and L-type (Figure 3F) channels. For P/Q-type channels, the distribution decreased compared to the control (left shift, Figure 3D), but for L-type channels, it increased (right shift, Figure 3F). For N-type channels, the distribution remained almost equal (Figure 3E). Thus, the presynaptic Ca^2+^ transients in neurons lacking Nxs are not only smaller as described already earlier [17,46], but also reveal a shift from vesicle-release-supporting VGCC subtypes P/Q- and N-type channels to Ca^2+^ channels that are primarily involved in slower processes like hormone secretion and Ca^2+^ signaling to gene transcription [11]. Both effects contribute to the weakening of synaptic transmission in neurons lacking Nxs [17,23,43,46].

### 3.4. Deletion of the Single Nx1α Variant Altered the Pattern of VGCC Subtype Contribution to Presynaptic Ca^2+^ Influx

Our measurements of total presynaptic Ca^2+^ transients revealed a smaller but significant impairment in neurons lacking only one Nx variant, Nx1α (Figure 2E,F). To investigate a possible specific impact of Nx1α on VGCC subtype distribution, which may differ from the changes caused by a lack of all Nx variants, we compared Cre^mut^- and Cre-transduced neurons from the new Nx1α cKO mouse line in terms of presynaptic VGCC subtype composition. Blocking the P/Q-type channel with ω-agatoxin IVA in these neurons induced an equal reduction in Ca^2+^ influx in control and Nx1α-deficient neurons (Figure 4A). Thus, the isolated P/Q-type Ca^2+^ transient is not affected by Nx1α and remains approximately at the same level (Nx1α Cre^mut^: 0.220 ± 0.045 vs. Nx1α Cre: 0.202 ± 0.40; Figure 4B). In contrast, the addition of ω-conotoxin GVIA resulted in a higher reduction in the Ca^2+^ influx in Nx1α-depleted cells and thus larger isolated Ca^2+^ influx through the N-type channels (Nx1α Cre^mut^: 0.104 ± 0.023 vs. Nx1α Cre: 0.158 ± 0.034; Figure 4B). After the addition of nifedipine, a higher Ca^2+^ influx remained in the Nx1α-depleted cells. Thus, the absolute influx through the L-type channels was reduced (Nx1α Cre^mut^: 0.146 ± 0.040 vs. Nx1α Cre: 0.069 ± 0.020; Figure 4B). The minor R-type transient remained almost unchanged, hardly above the noise level.

The evaluation of the relative portion of the different VGCC subtypes revealed a comparable result (Figure 4C). The cKO of Nx1α did not change the relative part of the P/Q-type channel (Nx1α Cre^mut^: 43.2 ± 1.3% vs. Nx1α Cre: 42.3 ± 1.4%, *p* > 0.9999). In contrast, a shift in the relative VGCC subtype contribution was seen in the N- and L-type channel contributions. The N-type portion significantly increased from 20.2 ± 0.8% (Nx1α Cre^mut^) to 31.2 ± 1.2% (Nx1α Cre; *p* < 0.0001); in contrast, the L-type portion decreased from 23.3 ± 1.0% (Nx1α Cre^mut^) to 8.0 ± 0.7% (Nx1α Cre; *p* < 0.0001). No significant changes were found in the R-type portion (*p* > 0.9999; Figure 4C). In summary, the N-type contribution increased whereas the L-type contribution decreased in neurons lacking Nx1α.

Again, we used the possibility to evaluate the VGCC contributions for each synaptic bouton individually and plotted a frequency histogram for P/Q-, N-, and L-type Ca^2+^ channels (Figure 4D–F). In a direct comparison between control cells and Nx1α-depleted cells, it appears that the Nx1α deletion led to a higher influx through the N-type channel as the peak at 10%, meaning only minor amounts of N-type channels in this bouton in control cells disappeared in neurons without Nx1α and the number of boutons with an N-type contribution above 30% was always moderately higher (Figure 4Ε). The distribution of L-type channel transients in single boutons showed that more boutons with a larger influx through the L-type channel existed under control conditions, and in the absence of Nx1α, more boutons were represented with almost zero Ca^2+^ influx through the L-type channels (Figure 4F). In summary, in neurons lacking Nx1α, the N-type contribution increased whereas the L-type contribution decreased and the P/Q-type was not affected; thus, the lack of the single Nx variant Nx1α led to significantly more vesicle-release-supporting N-type channels, which may compensate the effect of reduced total Ca^2+^ influx on vesicle release.

In combination, these results show that Nxs have an impact on the combination of presynaptic VGCC subtypes, but beyond this, it seems that individual Nx subtypes have correlations to special VGCC subtypes. Nx1α, which is prominently expressed in hippocampal neurons [48], supports L-type Ca^2+^ channels but seems to dampen the N-type Ca^2+^ channels, whereas the full Nx KO shows that all Nxs in concert promote the activity of P/Q- and N-type Ca^2+^ channels and, thus, presynaptic vesicle release.

### 3.5. Deletions of Nxs Also Affect the Endocannabinoid-Receptor-Dependent Modulation of Presynaptic Ca^2+^ Influx

Presynaptic VGCCs are modulated by a wealth of metabotropic receptors including the cannabinoid receptor CB1 [61,62]. Strikingly, the endocannabinoid receptor system was recently shown to be modulated by neurexins [44]. Endocannabinoids such as 2-arachidonoylglycerol (2-AG) are lipid-based neurotransmitters that bind to CB1R [63] and thereby allow the retrograde adaptation of synaptic activity [64,65,66,67]. This modulatory process is regulated postsynaptically by on-demand synthesis and degradation of endocannabinoids [68]. Here, we tested the idea that the decrease in presynaptic Ca^2+^ transients induced by the CB1-receptor agonist 2-AG depends on Nxs. Direct measurements of AP-driven presynaptic Ca^2+^ transients and subsequent activation of the CB1-receptor with 2-AG resulted in a significant reduction in Ca^2+^ influx in both control and Nx-deficient neurons as well as in β-Nx-deficient neurons (Figure 5). This reduction in the presynaptic Ca^2+^ influx by 2-AG was larger in control neurons (Nx123 cKO Cre^mut^: 0.37 ± 0.01; with 2-AG: 0.22 ± 0.01) compared to Nx-deficient neurons (Nx123 cKO Cre: 0.33 ± 0.01; with 2-AG: 0.24 ± 0.01; Figure 5E). The relative change (%) caused by the endocannabinoid 2-AG was significantly lower in the Nx-deficient neurons (Cre^mut^ −41.3% ± 1.0% vs. Cre −29.4% ± 1.1%, *p* < 0.0001; Figure 5F, blue columns), indicating a modulatory role of Nxs in the CB1-receptor signaling cascade. In neurons lacking only β-Nxs, the relative change caused by the addition of 2-AG was much weaker, but still significant (Cre^mut^: −41.3% ± 1.1% vs. Cre: −38.3% ± 1.0%; *p* = 0.047; Figure 5F, red columns). These data suggest a dominant role of α-Nxs in the Nx-related impact on retrograde endocannabinoid signaling, extending an earlier study describing a dependence solely on β-Nxs [44].

## 4. Discussion

The present study revealed an unexpectedly complex modulation of presynaptic Ca^2+^ influx by Nxs based on a comparison of Ca^2+^ transients through specific VGCC subtypes. In hippocampal neurons of Nx123 cKO mice, the presynaptic Ca^2+^ influx was reduced upon conditional knockout of all Nx variants (Figure 2D,F). Interestingly, this reduction was stronger than that previously seen in neurons lacking all α-Nxs but not β-Nxs [17] or in neurons lacking all β-Nxs but not α-Nxs [44,45]. But even our novel deletion of the ASD candidate gene Nx1α alone induced a reduced total presynaptic Ca^2+^ influx (Figure 2E,F), suggesting that already the lack of a single Nx variant affects synaptic efficiency. Obviously, the removal of more and more Nx variants gradually induces a stronger reduction in total presynaptic Ca^2+^ influx. While these data confirm our initial hypothesis of a general dose effect of Nxs on synaptic function [23], we surprisingly found that deletions of Nxs may produce different and complex patterns of affected VGCC subtypes.

In the complete Nx123 cKO, the reduced total presynaptic Ca^2+^ influx was mainly due to a reduced influx through P/Q-type channels (Figure 3). This is in line with similar data from hippocampal neurons of a constitutive knockout of all α-Nx variants [17] and from an analysis of the calyx of Held synapses, in which Nxs were shown to be crucial for clustering of P/Q-type channels in the active zone [47]. In addition, our investigation of Nx123 cKO produced a tendency toward the reduction in N-type channel contribution while increasing the contribution of L-type and R-type VGCCs. Since this shift would imply a change from channels directly coupled to vesicle release to channel subtypes with a mere supportive role in fast synaptic release [1,6], the transition away from P/Q- and N-type to L- and R-type channels likely predicts a more dramatic influence on synaptic release than the moderate reduction in the Ca^2+^ transients suggests. In fact, a large release defect has been previously described with a reduction in postsynaptic EPSCs in αTKO neurons (lacking all α-Nxs) by more than 50% compared to controls [42].

A different pattern of VGCC modulation was seen in the case of the single Nx1α cKO. The deletion of Nx1α alone did not change the Ca^2+^ influx through the P/Q-type channel but, unexpectedly, elevated the contribution of N-type channels. Since the total Ca^2+^ influx was moderately reduced in this deletion model, the increased Ca^2+^ influx through N-type channels was likely compensated by a reduced L-type channel contribution (Figure 4). As P/Q-type and N-type VGCCs are the main Ca^2+^ channels for presynaptic transmitter release and the deletion of Nx1α induces a shift in the relative contribution from L-type to N-type channels, transmitter release in Nx1α-deficient synapses should be normal or the probability of release should be increased, unlike in the complete deletion of Nxs. Thus, the overall organization of the presynaptic active zone and clustering of P/Q-type channels observed earlier in the calyx of Held synapses in the absence of all Nxs [47] most likely does not depend on the Nx1α variant because Nx1α knockout did not affect P/Q-type channel-driven Ca^2+^ influx as shown here in hippocampal neurons. However, this finding is in contrast to observations in neurons from a constitutive knockout of all α-Nx variants, in which overexpressed Nx1α partially rescued the amount of Ca^2+^ influx through P/Q-type channels [17]. This discrepancy indicates that Nx1α is not alone responsible for the modulation of the P/Q-type channel [17], but that other α-Nx variants can compensate for the deletion, e.g., in concert with α2δ auxiliary subunits of VGCCs. Together, these results are consistent with the view that Nxs regulate presynaptic Ca^2+^ influx and that individual Nx variants may have partially overlapping, partially non-redundant effects on the distribution or function of different VGCC subtypes.

To further explore the possibility that α-Nxs are also involved in additional signaling pathways targeting presynaptic VGCCs as suggested previously for the GABA_B_ receptor pathway [43,69], we investigated retrograde signaling via the endocannabinoid system (ECS). In this retrograde pathway, postsynaptically synthesized endocannabinoids (e.g., 2-AG or AEA) diffuse to the presynapses to stimulate presynaptic CB1-receptors, inhibiting the activity of VGCCs [64,65]. We therefore compared neurons containing or lacking Nxs and revealed a lower relative CB1-receptor effect by 2-AG in the absence of all Nx variants. This is in line with an increased tonic endocannabinoid signaling as has been proposed for reduced β-Nx levels according to a study of the ECS in β-Nx-deficient neurons [44]. Our current results now suggest that the role of α-Nxs in this regulation may even be stronger than that of β-Nxs, based on a direct comparison of neurons lacking all Nx123 cKO versus β-Nx cKO neurons. While these results present an important extension of the role of Nxs in regulating presynaptic VGCCs, the measured changes in presynaptic Ca^2+^ transients could not elucidate the precise mechanism of how Nxs modulate the ECS. However, at least three hypotheses are conceivable. First, a postsynaptic regulation of 2-AG synthesis as postulated in [44] is possible since Nxs engage in transsynaptic interactions and can cluster receptors in the postsynaptic membrane, for example, AMPAR [48,70,71] and GABA_A_R [72]. Naturally, such a potential postsynaptic influence on 2-AG synthesis could hardly be attributed to β-Nxs alone as α-Nxs share the same binding partners, supporting our observation here. This scenario would imply that α-Nxs have an additional effect on VGCCs by modulating the ECS postsynaptically. Second, Nxs could modulate the effect of the ECS system via the presynaptic organization of VGCCs and/or the localization of CB1-receptors. In support, it was shown that α-Nxs are presumably involved in the overall organization at the active zone [47], and an altered distribution or activity of either CB1-receptors or the VGCC subtypes themselves may explain the effect of Nxs reported here. Third, since the activity of the CB1-receptor is regulated by on-demand production and degradation of 2-AG [68], it cannot be ruled out that Nxs might influence presynaptic 2-AG degradation as an additional alternative. Future research will have to distinguish between these possibilities.

## 5. Conclusions

Neurexins are key players in synapse organization and were found here to be pivotal for the contribution of particular Ca^2+^ channel subtypes to presynaptic Ca^2+^ influx into boutons of cultured hippocampal neurons. The lack of all Nx isoforms weakened the contribution of the P/Q-type channels that are normally responsible for fast synaptic vesicle release and elevated the amount of Ca^2+^ influx via L-type channels. In contrast, the deletion of the single Nx1α variant alone promoted influx through N-type channels at the expense of L-type channels. Our data indicate a complex interplay of different Nx variants with several subtypes of synaptic Ca^2+^ channels. This complex interplay may include the modulation of the endocannabinoid system by α-Nxs that also impacts synaptic Ca^2+^ channels.

## Figures and Tables

**Figure 1 cells-13-00981-f001:**
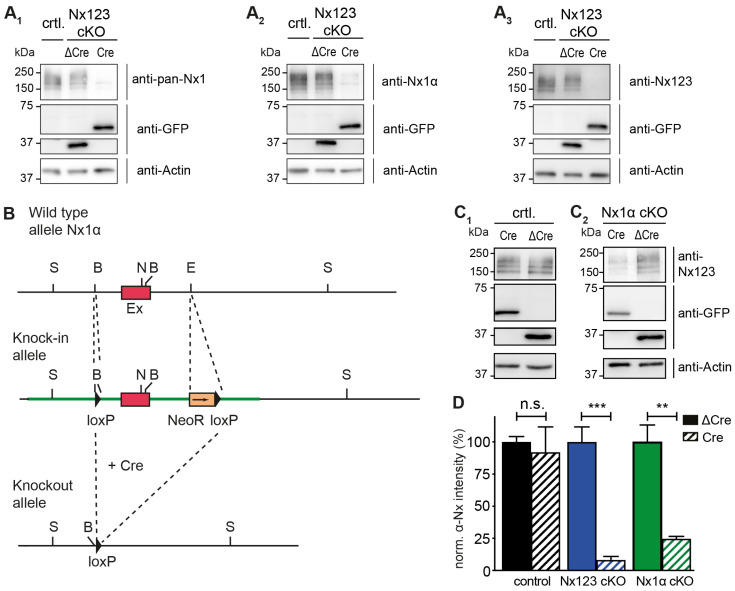
Conditional deletion of the single Nx1α variant. (**A**) Immunoblots of Nxs in Nx123 cKO cells tested with 3 different antibodies: pan-Nx1 (**A_1_**, Millipore #AB161-I), Nx1α (**A_2_**, Frontier Institute #AB_2571817), and Nx123 (**A_3_**, SySy #175003). Con, mouse line without floxed α-Nxs. (**B**) Wild-type allele of the 5′ end of the Nx1α gene including the first coding exon (indicated in red) is illustrated. After successful homologous recombination of the wild-type allele with the targeting vector (not depicted), the knock-in allele that resulted is indicated. The 5′ loxP site is introduced via the BamH1 (‘B’) site upstream of the first coding exon. Downstream of the first coding exon and at the EcoR1 site (‘E’), the 3′ loxP site and NeoR (Neomycin resistance) gene are inserted (blunt-end cloning). Via the addition of a Cre-recombinase, the knock-in allele is converted into the knockout allele. The region between the loxP sites of the Nx1α gene including the first coding exon is excised. Further restriction sites: S = Spel, N = Nhel. (**C**) Immunoblots of control neurons (**C_1_**) without floxed α-Nx and Nx1α cKO neurons (**C_2_**) were probed with anti-Nx123 (SySy #175003). (**D**) Quantification of αNx normalized to ΔCre condition (100%) for control neurons, Nx123 cKO neurons, and Nx1α cKO neurons. Data are based on n independent immunoblot experiments like in **A_3_** and **C** (control: 3, 2; Nx123: 4, 4; Nx1α: 3, 3); columns were compared with an unpaired *t*-test. n.s. = non-significant: *p* > 0.05, ** *p* < 0.01, *** *p* < 0.001.

**Figure 3 cells-13-00981-f003:**
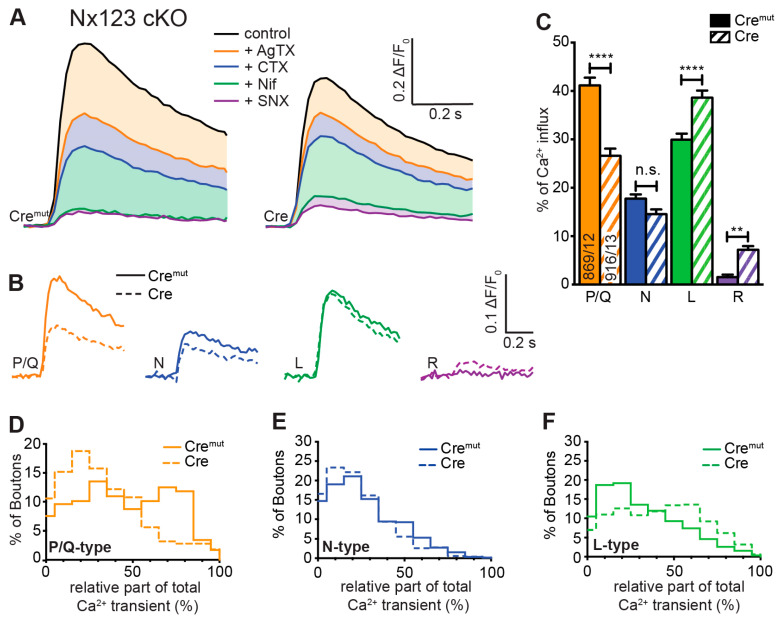
Deletion of all Nx variants decreased presynaptic Ca^2+^ influx primarily via P/Q-type VGCCs. Pharmacologically isolated VGCC subtype contribution to the Ca^2+^ influx measured during single-AP stimulation in Nx123 cKO neurons with synGCaMP7b by sequential addition of specific blockers: ω-agatoxin IVA (AgTX, 0.1 μM; P/Q-type); ω-conotoxin GVIA (CTX, 2 μM; N-type); nifedipine (Nif, 20 μM; L-type); SNX-482 (SNX, 0.5 μM; R-type). (**A**) Averaged traces of control neurons (Cre^mut^, 12 cells/869 boutons, left) and neurons lacking all neurexin variants (Cre, 13/916, right); colors indicate traces *after* subsequent application of subtype-specific blockers as depicted. Thus, the area in dimmed colors above the traces indicates the amount of Ca^2+^ influx sensitive to the given blocker. (**B**) Ca^2+^ transients that reflect Ca^2+^ influx through the given VGCC subtypes are isolated by subtraction from the traces in A, comparing Nx123 cKO Cre^mut^ (continuous lines) and Nx123 cKO Cre (dashed lines). (**C**) Mean ± SEM of relative VGCC subtype contribution (% of control) calculated for each bouton (ROI, relative to total Ca^2+^ influx) in Nx123 cKO neurons. The number of examined boutons/cells is shown in the P/Q columns and applies to all VGCC subtypes. Columns were compared with Kruskal-Wallis test, n.s.: *p* > 0.05, **: *p* < 0.01, ****: *p* < 0.0001. (**D**) Each presynaptic bouton’s P/Q-type contribution was determined, and the spreading is depicted in a histogram that contrasts the distribution of Nx123 cKO neurons with and without Nxs. The data show that without Nxs, the number of boutons with more than 50% P/Q-type Ca^2+^ influx is almost lost (Cre^mut^, 584/12; Cre, 501/13). The same analysis is shown in (**E**) for the N-type and (**F**) for the L-type portion in individual boutons.

**Figure 4 cells-13-00981-f004:**
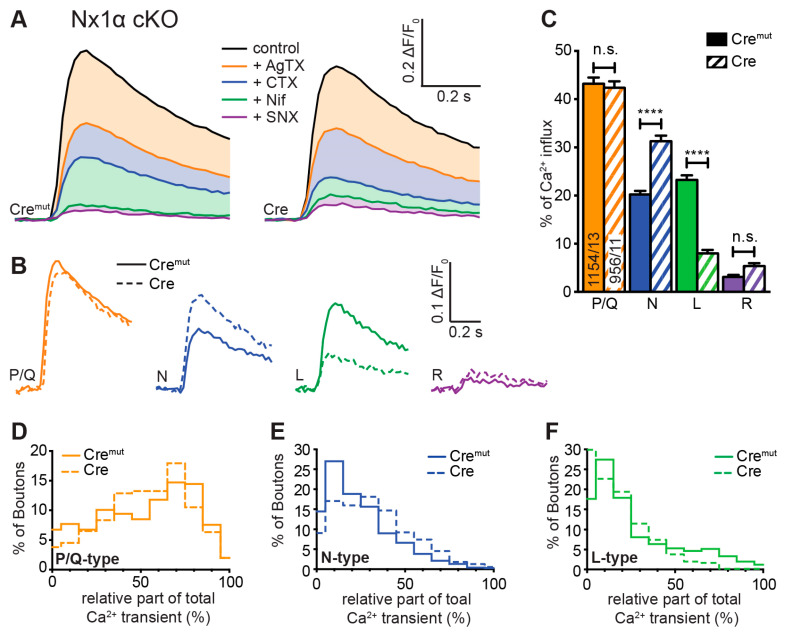
Single Nx1α deletion changed the VGCC subtype contribution to presynaptic Ca^2+^ influx. VGCC subtype contribution to Ca^2+^ influx after single-AP stimulation was measured in Nx1α cKO neurons with synGCaMP7b by sequential addition of specific blockers as described in Figure 3. (**A**) Averaged traces of control neurons (Nx1α cKO Cre^mut^, 13 cells/1154 boutons; *left*) and neurons lacking only Nx1α (Cre, 11/956, *right*); colors indicate traces *after* subsequent application of subtype-specific blockers as depicted. Thus, the area in dimmed colors above the traces indicates the amount of Ca^2+^ influx sensitive to the given blocker. (**B**) Ca^2+^ transients that specifically reflect Ca^2+^ influx through the sequentially blocked VGCC subtypes were isolated by subtraction from the traces in A and compared between Nx1α cKO Cre^mut^ (continuous lines) and Nx1α cKO Cre (dashed lines). (**C**) Mean ± SEM of relative Ca^2+^ contribution (% of control) per VGCC subtype calculated for individual boutons (ROIs) in Nx1α cKO neurons. The number of examined boutons/cells is shown in the P/Q columns and applies to all VGCC subtypes. Columns were compared with Kruskal-Wallis test, n.s.: *p* > 0.05, ****: *p* < 0.0001. (**D**) The P/Q-type portion of Ca^2+^ transients was calculated for each synaptic bouton, and the spreading is shown in a histogram comparing the variability in neurons with and without Nx1α (Cre^mut^, 755/13; Cre, 552/11), in (**E**) for the N-type and (**F**) for the L-type.

**Figure 5 cells-13-00981-f005:**
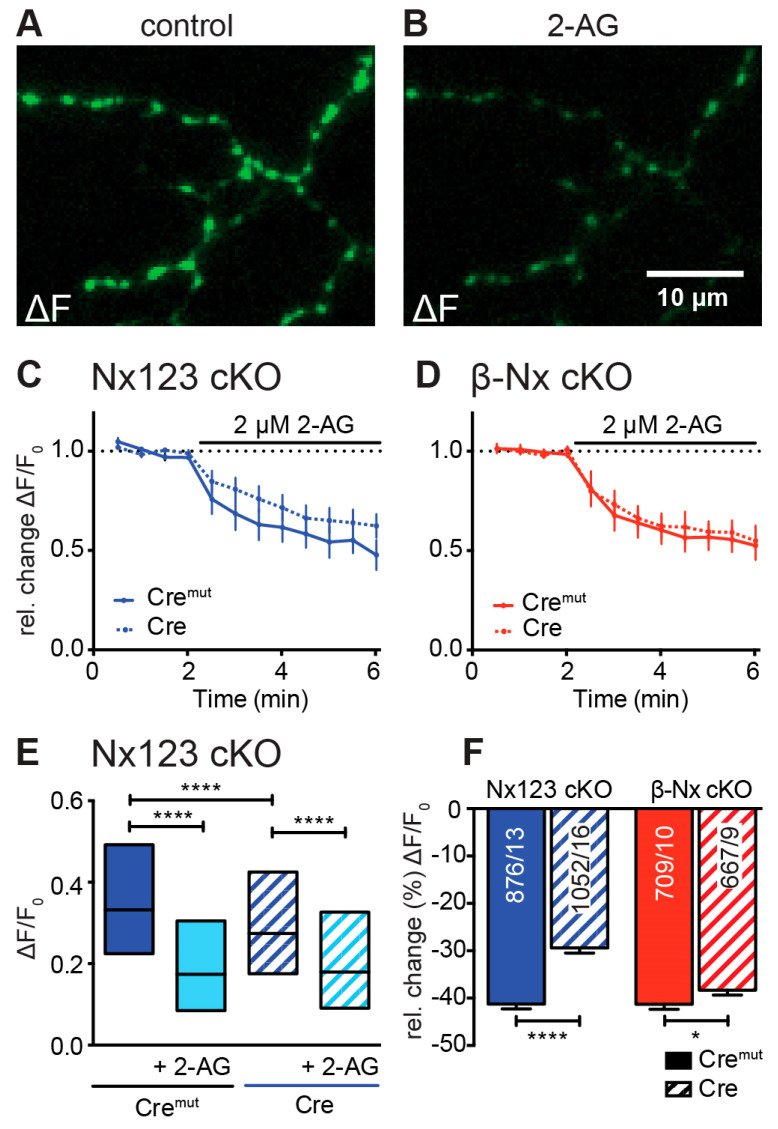
Endocannabinoid-evoked CB1-receptor activation reduces presynaptic Ca^2+^ transients in an Nx-dependent manner. (**A**) Several presynaptic boutons of a synGCaMP7b-transfected Nx123 cKO Cre neuron are shown in an exemplary ΔF image during 3-AP stimulation. (**B**) The identical presynaptic boutons to those in A after 5 min of CB1-receptor activation with 2 µM 2-AG, again during a 3-AP stimulation. (**C**) Repetitive stimulation (1 AP every 30 s) shows a reduction in Ca^2+^ transients in response to the application of 2-AG, averaged (mean ± SEM) from neurons of Nx123 Cre^mut^ (13 neurons, continuous line) and Nx123 Cre (16, dotted line), displayed as relative changes normalized to the mean of four stimulations before 2-AG application. (**D**) Similar recordings as in C for β-Nx cKO Cre^mut^ (10, continuous line) and β-Nx cKO Cre (9, dotted line). (**E**) Boxplot (quartiles and median) of Ca^2+^ ΔF/F_0_ for Nx123 Cre^mut^ (960 ROIs/13 cells) and Nx123 Cre (1168/16) before and after 2-AG application. (**F**) Relative change (%) in presynaptic Ca^2+^ transients by activation of CB1-receptor with 2-AG. Cells were measured under both conditions (control and 5 min of 2-AG), and reduction was calculated for each bouton separately, plotted as mean ± SEM in Nx123 cKO (blue) and β-Nx cKO (red) neurons (Cre^mut^ and Cre). Outliers were detected and removed with the ROUT method (Q = 1), and columns were compared with an unpaired *t*-test; * *p* < 0.05; **** *p* < 0.0001; numbers (included ROIs/cells) are given in the columns.

## Data Availability

The raw data supporting the conclusions of this article will be made available by the authors on request.

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
