# Peer review of "Conditional Knockout of Neurexins Alters the Contribution of Calcium Channel Subtypes to Presynaptic Ca2+ Influx"

_cells, 2024, doi:10.3390/cells13110981_

Round 1

Reviewer 1 Report

Comments and Suggestions for Authors

The study by Brockhaus et.al investigates the role of Nx variants in regulating presynaptic Ca2+ influx in hippocampal neurons. Indeed, this is the continuation of their previous studies. They have found that the deletion of all Nx variants or specifically the Nx1α variant led to a substantial decrease in α-Nx expression and presynaptic calcium transients. This pivotal finding underscores that different neurexin variants may have distinct impacts on specific VGCCs, highlighting the complexity and specificity of neurexin function in neuronal signaling.

Major points:

In Fig. 3A and 4A, Ca2+ transients that specifically reflect Ca2+ influx through L-type (green color) look similar (a bit higher in both in Nx123cKO and Nxα1cKO) but in Fig. 3C, it looks increased and in 4C decreased. Could authors explain the calculation?

Minor points:

1.       Fig 1. Author should explain WT and αNX WT (Fig. 1A and C) in the legend and is there any difference in it.  Fig. 1 C a bit confusing with two panels.

2.       I would also suggest that author should check Nx1α expression in Nx1α cKO neurons at they have antibody already (Shown in A2)

3.       Fig 1 legend. provide more info in (D). I assume this is quantification from A and C wb image. Please mention it.

4.       Suggestion for Fig.2 D, E, F. Make it only D and E. D transient and graph of Nx123 cKO and E for transient and graph ofNx1α cKO.

Did the author also check how deletions of Nx1α affect the endocannabinoid receptor-dependent modulation of presynaptic Ca2+ Influx. It should be mentioned as this paper is about the variant of Nx.

Reviewer 2 Report

Comments and Suggestions for Authors

The paper entitled “Conditional Knockout of Neurexins Alters the Contribution of 2 Calcium Channel Subtypes to Presynaptic Ca2+-Influx” by Brockhaus and colleagues aim to investigate the alteration in presynaptic calcium influx in the neurons deleted for several variants of neurexin. The paper is scientifically sound and highlight the complexity of calcium regulation at boutons in the presynaptic neuron. In particular, the contribution of the several Nx variants in the regulation of calcium influx mediated by VGCCs is well dissected. The work is simple, but linear and complete. Thus, in my opinion it deserves to be published. However, I have some little questions.

1. Why the authors choose to infected cells with the Ca2+ indicator synGCaMP7b instead to use probes like Fura-2AM or similar?

2. Did the authors investigate if the reduce calcium influx consequently caused an altered calcium release from the storages?

3. It is not so clear why the authors have chosen to investigate the CB1R among all the metabotrobic receptors modulating presynaptic VGCCs. Could you kindly better explain it?

4. I think that the link with ASD (last paragraph of the discussion) is too much forced. I would remove this paragraph. On the contrary, the authors have to better contextualize their data and the ASD, introducing the pathology in the introduction and explaining the relapse of their research in autism field.

Round 2

Reviewer 1 Report

Comments and Suggestions for Authors

You can write "control" as "ctrl" in figures instead of using "Con." "Ctrl" is widely used.